# Music-Based Intervention Ameliorates *Mecp2*-Loss-Mediated Sociability Repression in Mice through the Prefrontal Cortex *FNDC5/BDNF* Pathway

**DOI:** 10.3390/ijms22137174

**Published:** 2021-07-02

**Authors:** Pi-Lien Hung, Kay L. H. Wu, Chih-Jen Chen, Ka-Kit Siu, Yi-Jung Hsin, Liang-Jen Wang, Feng-Sheng Wang

**Affiliations:** 1Department of Pediatrics, Kaohsiung Chang Gung Memorial Hospital, College of Medicine, Chang Gung University, Kaohsiung 83301, Taiwan; flora1402@cgmh.org.tw (P.-L.H.); superarthy2@gmail.com (C.-J.C.); 2Center for Translational Research in Biomedical Sciences, College of Medicine, Kaohsiung Chang Gung Memorial Hospital, Chang Gung University, Kaohsiung 83301, Taiwan; klhwu@cgmh.org.tw; 3Department of Orthopedic Surgery, Kaohsiung Chang Gung Memorial Hospital, College of Medicine, Chang Gung University, Kaohsiung 83301, Taiwan; michaelskk@gmail.com; 4Department of Physical Medicine and Rehabilitation, Kaohsiung Chang Gung Memorial Hospital, Kaohsiung 83301, Taiwan; yjhsin@cgmh.org.tw; 5Department of Child and Adolescent Psychiatry, Kaohsiung Chang Gung Memorial Hospital, Kaohsiung 83301, Taiwan; anus78@cgmh.org.tw; 6Core Facility for Phenomics and Diagnostics, Kaohsiung Chang Gung Memorial Hospital, Kaohsiung 83301, Taiwan; 7Center for Mitochondrial Research and Medicine, Kaohsiung Chang Gung Memorial Hospital, Kaohsiung 83301, Taiwan; 8Department of Medical Research, Kaohsiung Chang Gung Memorial Hospital, College of Medicine, Chang Gung University, Kaohsiung 83301, Taiwan

**Keywords:** *mecp2* ^null^^/y^ mice, music-based intervention, brain-derived neurotrophic factor (*BDNF*), *FNDC5* gene

## Abstract

Patients with Rett syndrome (RTT) show severe difficulties with communication, social withdrawl, and learning. Music-based interventions improve social interaction, communication skills, eye contact, and physical skills and reduce seizure frequency in patients with RTT. This study aimed to investigate the mechanism by which music-based interventions compromise sociability impairments in m*ecp2* ^null/y^ mice as an experimental RTT model. Male m*ecp2* ^null/y^ mice and wild-type mice (24 days old) were randomly divided into control, noise, and music-based intervention groups. Mice were exposed to music or noise for 6 h/day for 3 consecutive weeks. Behavioral patterns, including anxiety, spontaneous exploration, and sociability, were characterized using open-field and three-chamber tests. *BDNF*, TrkB receptor motif, and *FNDC5* expression in the prefrontal cortex (PFC), hippocampus, basal ganglia, and amygdala were probed using RT-PCR or immunoblotting. *m**ecp2* ^null/y^ mice showed less locomotion in an open field than wild-type mice. The social novelty rather than the sociability of these animals increased following a music-based intervention, suggesting that music influenced the m*ecp2*-deletion-induced social interaction repression rather than motor deficit. Mechanically, the loss of *BDNF* signaling in the prefrontal cortex and hippocampal regions, but not in the basal ganglia and amygdala, was compromised following the music-based intervention in m*ecp2* ^null/y^ mice, whereas TrkB signaling was not significantly changed in either region. *FNDC5* expression in the prefrontal cortex region in m*ecp2* ^null/y^ mice also increased following the music-based intervention. Collective evidence reveals that music-based interventions improve m*ecp2*-loss-induced social dysfunction. *BDNF* and *FNDC5* signaling in the prefrontal cortex region mediates the music-based-intervention promotion of social interactions. This study gives new insight into the mechanisms underlying the improvement of social behaviors in mice suffering from experimental Rett syndrome following a music-based intervention.

## 1. Introduction

Rett syndrome (RTT) is a neurodevelopmental disorder characterized by severe developmental delay, an autistic-like phenotype, inappropriate behaviors, a labile mood, social withdrawal, epilepsy, and learning disabilities [1,2]. Patients with RTT exhibit a loss of hand function skills during a regression period [3] and a slow decline in gross motor skills over time [4]. *Mecp2* gene mutation is one of the prominent genetic features of RTT development [5,6]. While the assessment of potential intellectual alterations in patients with RTT remains uncertain, special education strategies are highly recommended. Studies have shown that providing patients with RTT with a strong motivational factor can promote or motivate them to communicate or learn [7]. Of these strategies, music-based interventions have been recommended to motivate these patients to learn [6]. Case studies have revealed that music-based interventions promote patients’ social interactions and communication and improve their cognition and affective/physical skills [8,9,10,11,12,13]. We previously showed that the social interactions, communication skills, eye contact, hand function, and seizure frequency of patients, in addition to parenting stress, improved following a 24-week music-based intervention [14]. It has been found that music-based interventions are effective for treating some neuropsychiatric disorders, including dementia [15] and autism [16]. However, the molecular mechanism by which music-based interventions promote social behavior remains uncertain.

RTT and autistic spectrum disorder (ASD) are neurodevelopmental disorders that share common features, including limited eye contact and poor socialization; however, clear differences exist between the two disorders in the aspects of clinical presentation and neurotrophin signaling. As for clinical considerations, autism predominantly occurs in males, and it usually does not involve genetic mutations. ASD is associated with an accelerated head growth rate. In addition, ASD has a greater incidence that RTT (1:100 vs. 1:1500 female births). RTT is a X-linked disorder involving the m*ecp2* gene mutation, so females are mostly affected. RTT is associated with a deceleration in the rate of head growth postnatally. As for the molecular mechanism responsible for neurotrophin signaling, increasing evidence shows that *mecp2* loss induces underproduction of brain-derived neurotrophic factor (*BDNF*), suppressing brain development and function [17]. *BDNF* plays an important role in the development of RTT [18,19,20]. Patients with ASD were shown to have a significantly elevated *BDNF* level as well, and elevated *BDNF* has also been associated with autistic traits in the general population [21]. In this study, we investigate how music-based interventions affect RTT, a condition that presents a paradoxically different *BDNF* level from what is seen in ASD.

Previous studies have demonstrated that the levels of *BDNF*, NGF elevated [22], and some genes were regulated after music exposure [23]. Environmental enrichment with a music-based intervention or sensory stimulation can increase the serum *BDNF* level and improve social interactions in patients with early stage Rett syndrome [24]. In addition, this molecule mediates music-induced synaptic plasticity [20,21,22,25,26,27]. Music-based interventions activate a plethora of specific pathways in different brain regions including the cingulate gyrus, hippocampus, hypothalamus, hypothalamus, amygdala, and prefrontal cortex, thereby changing human emotional behaviors. In experimental animals, music has been shown to increase the activity of the BNDF/TrkB signaling pathway in the hypothalamus or hippocampus regions, reducing anxiety-related behavior and improving learning performance [22,28,29].

This study used *mecp2* ^null^ mice as an experimental RTT model to investigate whether music exposure affects anxiety-like behavior and/or reciprocal social interactions and to probe whether the *BDNF* or TrKB signaling pathways in brain regions involving emotional control, including the prefrontal cortex, hippocampus, amygdala, and basal ganglia, are affected by a music-based intervention.

## 2. Results

### 2.1. Mecp2 ^null/y^ Mice Show a Motor Deficit Rather Than Anxiety

*Mecp2* ^null/y^ mice showed a genotype lacking the *mecp2* gene, corresponding to a region of 465 bp, in agarose electrophoretograms (Figure 1A). Motor deficits and anxiety-related behaviors in experimental animals were assessed by an open field test. When exposed to the novel environment in the open field arena, WT mice engaged more in exploratory behaviors, especially in the central zone, than *mecp2*
^null/y^ mice. In addition, WT mice tracked crosses into the center of the arena at regular intervals, while the *m**ecp2 ^null/y^* mice remained in proximity to the walls of the arena, indicating a tendency toward anxiety-related behaviors in these animals (Figure 1B). The total distance traveled in the arena was significantly shorter for the C-*mecp2*
^null/y^ mice and W-*mecp2*
^null/y^ mice as compared with the WT mice. *mecp2* ^null/y^ mice did not show significant motor deficits as compared with other groups (Figure 1C, *p* < 0.05). *Mecp2* ^null/y^ mice tended to spend less time in the central field. However, this was only observed for several experimental mice, and the results did not reach statistical significance (Figure 1D). The results indicate motor dysfunction in these animals. The music-based intervention did not significantly affect the moving distance or time in WT or *mecp2*
^null/y^ mice.

### 2.2. Music-Based Interventions Can Reduce Social Deficits in Mecp2^null/y^ Mice

We performed a social novelty preference test to investigate whether animals’ social behaviors were affected by a music-based intervention. A decrease in sniffing of stranger mice in Holding Zone 1 indicates an impairment in sociability parameters. While the three groups of *mecp2 ^null/y^* mice did not show significant differences in sniffing behaviors compared with the three groups of WT mice (Figure 1E), *mecp2 ^null/y^* mice in the music group spent more time sniffing stranger 2 in Holding Zone 2 than stranger 1 in Holding Zone 1. A similar phenomenon was observed for WT mice in the control and white noise groups (Figure 1F, *p* < 0.05). The result indicates that the music-based intervention improved one of the social parameters, sniffing intensity, in *mecp2 ^null/y^* mice.

### 2.3. The Music-Based Intervention Altered BDNF mRNA and Protein Levels in Different Brain Regions

The analysis of improvements in social interactions in *mecp2 ^null/y^* mice following a music-based intervention prompted us to investigate the affects on *BDNF* signaling in particular brain regions. *BDNF* mRNA expression was significantly decreased in the prefrontal cortex region in the control *mecp2 ^null/y^* and white-noise-treated *mecp2 ^null/y^* mice as compared with the control WT and white-noise-treated WT mice (Figure 2A, C-*mecp2 ^null/y^* and W-*mecp2 ^null/y^* vs. C-WT and W-WT, *p* < 0.05). An intragroup comparison of *BDNF* mRNA expression in the prefrontal cortex region showed a significant increase in *BDNF* mRNA in *mecp2 ^null/y^* mice and music-treated *mecp2 ^null/y^* mice as compared with control *mecp2 ^null/y^* mice (Figure 2A, M-*mecp2 ^null/y^* vs. C-*mecp2 ^null/y^*, *p* < 0.05). In the hippocampal region, *BDNF* mRNA expression was significantly decreased in white-noise-treated *mecp2 ^null/y^* mice as compared with WT mice and *mecp2 ^null/y^* mice in the other two groups (Figure 2B, W-*mecp2 ^null/y^* mice vs. WT and other groups of *mecp2 ^null/y^* mice, *p* < 0.05). Signaling in the basal ganglia (Figure 2C) and amygdala (Figure 2D) was not significantly altered in WT or *mecp2 ^null/y^* mice.

*BDNF* protein levels in the prefrontal cortex region were not significantly changed in any of the mcie (Figure 3A), whereas the protein was significantly increased in the hippocampi of music-treated *mecp2 ^null/y^* mice as compared with all WT mice and the other two groups of *mecp2 ^null/y^* mice (Figure 3B, M-*mecp2 ^null/y^* vs. all WT and the other two groups of *mecp2 ^null/y^* mice, *p* < 0.05). Consistent with the levels of *BDNF* mRNA expression in the basal ganglia and amygdala, the *BDNF* protein levels were not significantly affected in these two brain regions (Figure 3C,D). The analysis indicates that BNDF signaling in the brain microenvironment following a music-based intervention is context-dependent.

### 2.4. Mecp2 Deletion Rather Than Music-Based Intervention Affected the TrkB Receptor in the Prefrontal Cortex

TrkB, receptor tyrosin kinase, is an important downstream receptor of *BDNF* [30,31]. We investigated whether TrkB signaling in the prefrontal cortex or hippocampus changed following a music-based intervention. TrkB mRNA expression in the prefrontal cortex was significantly increased in the control and white-noise-treated *mecp2*
^null/y^ mice as compared with WT and *mecp2* ^null/y^ mice in the music group (Figure 4A, C-*mecp2*
^null/y^ and W-*mecp2*
^null/y^ mice vs. all WT and M-*mecp2*
^null/y^ mice, *p* < 0.05). Signaling in the hippocampus was not significantly changed in any of the mice (Figure 4B).

### 2.5. Mecp2 ^null/y^ Mice That Did Not Undergo a Music-Based Intervention Overproduced Full-Length and Truncated TrkB Protein in the PFC

TrkB contains spliced isoforms, including a full-length, catalytically active kinase receptor (TrkB-FL), and truncated isoforms (TrkB-T). TrkB-T and TrkB-FL have important interactions with neurotrophins to regulate neural activity [32]. We investigated whether TrkB-FL or TrkB-L signaling in the prefrontal cortex or hippocampus changed following music treatment. TrkB-FL and TrkB-T protein levels in the prefrontal cortex increased significantly in control C-*mecp2 ^null/y^* mice as compared with all WT mice (Figure 5A, C-*mecp2 ^null/y^* vs. all WT mice, *p* < 0.05); however, these molecules were not significantly affected in the hippocampal region in any of the mice (Figure 5B).

### 2.6. Music-Based Interventions Induce Prefrontal Cortex FNDC5 Gene Expression

The analysis of motor deficits in *mecp2 ^null/y^* mice prompted us to ask whether myokine *FDNC**5* was changed following a music-based intervention, as *FNDC5* is important for muscle function and cognition. The expression of *FNDC5* in the prefrontal cortex was significantly upregulated in music-treated *mecp2 ^null/y^* mice as compared with white-noise-treated *mecp2 ^null/y^* mice (Figure 6A, M-*mecp2 ^null/y^* vs. W-*mecp2 ^null/y^* mice, *p* < 0.05). Of note, the myokine was significantly reduced in all *mecp2 ^null/y^* mice as compared with all WT mice (Figure 6B, all *mecp2 ^null/y^* mice vs. all WT mice, *p* < 0.05).

## 3. Discussion

Our data provide evidence that music-based interventions can restore the social abilities of Rett syndrome (RTT)mice through the upregulation of *BDNF* mRNA expression in the prefrontal cortex and an increase in the *BDNF* protein level in the hippocampal region. In addition, our data reveal that music-based interventions can upregulate *BDNF* gene expression in a TrkB-pathway-independent manner through modulation of the *FNDC5* gene.

RTT is caused by mutations of the *mecp2* gene [5]. The spectrum of *mecp2* gene mutations has been found to involve missense, nonsense, and frame-shift mutations, as well as truncations due to premature STOP codons, with specific mutations correlating with the level of clinical severity [33,34]. The *mecp2* gene located in X chromosome Xq29 encodes the *mecp2* protein which binds to methylated CpG sites in the gene promotor and is critical for binding to regulatory gene regions and for the recruitment of cofactors. Close to 1300 genes will be transcriptionally dysregulated when the *mecp2* gene is dysfunctional. The first mammalian neuronal target gene for *mecp2* identified was brain-derived neurotrophic factor (*BDNF*) [35,36]. *Mecp2* dysfunction leads to reduced production of *BDNF*, a protein required for normal neuronal development. Significant evidence indicates a reduction in *BDNF* levels in *mecp2*-based mouse models of RTT, which becomes significant with the appearance of RTT-like features. Two studies showed lower *BDNF* mRNA levels in autopsy brain samples from RTT individuals [37,38], which is reminiscent of the situation in *mecp2* mutant mice. Improving *BDNF* expression and/or signaling has received a significant amount of attention regarding the treatment of a variety of neurological disorders [39,40], and a great deal of progress has been achieved by the RTT research community [41,42].

Music-based interventions have been used in some descriptive studies and case reports to restore communication, voluntary hand movement, and decreased stereotypic movement abilities [43,44]. A previous comprehensive study conducted by the current authors demonstrated that a music-based intervention improved receptive language, verbal and nonverbal communication skills, and social interactions in RTT patients. Purposeful hand functions, breathing patterns, and eye contact were significantly improved. In addition, the music-based intervention also decreased the frequency of epileptic seizures [14]. Previous research reported that wild-type adult mice or prenatal auditory stimulation can increase the *BDNF* protein level in the hippocampus [28,45,46,47] as well as in the PFC and amygdala [47]. It was also reported that the *BDNF*/TrkB level in the dorsal hippocampus CA3 (dCA3) and dentate gyrus (dDG) was significantly enhanced in rats exposed to Mozart’s music as compared with those without music exposure [25]. Corresponding to the results of previous studies, our data suggest that the music-based intervention upregulated *BDNF* mRNA expression in the PFC and increased *BDNF* protein levels in the hippocampal regions in *mecp2 ^null/y^* mice. However, neither full-length nor truncated TrkB receptors were upregulated in m*ecp2 ^null/y^* mice exposed to music. On the contrary, *mecp2 ^null/y^* mice that did not undergo a music-based intervention overexpressed TrkB-FL and TrkB-T receptors. We presume that impaired *BDNF* gene expression is the major pathophysiology of neurological dysfunction in RTT, and a *BDNF* protein secretion deficiency in the brain leads to compensation for TrkB receptor overexpression, either in full-length or truncated isoforms. *Mecp2 ^null/y^* mice exposed to music showed upregulated *BDNF* gene expression and relatively normalized TrkB levels. The precise function of TrkB-T receptors still remains elusive. However, previous studies reported that TrB-T binds to *BDNF* with similar affinity to the TrkB kinase and can interfere with *BDNF*-TrkB signaling by binding to *BDNF* without activating downstream kinase cascades [48,49]. This explains why the compensation for TrkB receptor overexpression in *mecp2 ^null/y^* mice with music exposure is not successful.

Fibronectin type III domain containing 5 (*FNDC5*) is known to induce *BDNF* expression in mice. Physical activity is an affordable and effective method that can be used to improve cognition by inducing *FNDC5* gene expression. After cleavage by protease in the skeletal muscle, *FNDC5* is cleaved into a 12 kDa peptide called irisin, which is able to enter the central nervous system and induce *BDNF* gene expression in the hippocampus. The finding that music-based intervention can induce *FNDC5* expression in the PFC of *mecp2^null/y^* mice is novel. We suggest that the upregulation of the *BDNF* gene after music-based intervention in the PFC of *mecp2 ^null/y^* mice in our study occurred through the modulation of *FNDC5* gene expression. The irisin immunoreactivity could not be investigated in this study since it has been reported to only exist in GABA-ergic purkinje cells in the cerebellum and vestibular nuclei of the medulla oblongata [50].

We successfully established a music-based intervention model in RTT in vivo; however, the analysis of *FNDC5*/*BDNF* signaling pathways hinted at the complex nature of biological activity in brain tissue that the intervention may influence. Engibeerin cortical organoids from human-induced pluripotent stem cells (iPSCs), showing neurite undergrowth, neurite coalescence, and soma size of interneurons, have recently been used as an in vitro RTT model [51,52]. The biological response or the roles of *FNDC5*/*BNDF* pathways in an in vitro humanized RTT model upon different music-based interventions, such as magnitude and frequency (Hz) of music, warrant further investigations in the future.

We acknowledge the limitations of this study. First, we used male *mecp2 ^null/y^* mice due to the rapid onset of the Rett-like phenotype in this species; however, the clinical relevance of Rett syndrome still remains doubtful. Second, there are several behavioral tests that can be used to assess whether mice show autistic-like social interaction deficits, and the three-chamber social approach is commonly used in mice. However, whether the three-chamber sociability test is the best choice for *mecp2 ^null/y^* mice remains elusive. Third, the music chosen for use in this study contained complicated in-music elements, such as rhythm, melody, harmony, timbre, voice pitch, etc. We cannot exclude the possibility that each element may have an impact on *mecp2 ^null/y^* mice. Fourth, a study has revealed that very little *FNDC5* mRNA is present in the hippocampus [50]. The biological function of hippocampal *FNDC5* to the RTT model merits analysis.

## 4. Materials and Methods

### 4.1. Animals

Experimental procedures followed ethical guidelines and were approved by the Institutional Animal Care and Use Committee, Kaohsiung Chang Gung Memorial Hospital (project identification code: CMRPG8G0961, date of approval: 20 March 2017). *Mecp2* ^null/y^ mice were purchased from Jackson Laboratory (Bar Harbor, ME, USA; strain name: *m**ecp2*
^tm1−1Bird^, stock number: 003890) and bred at the Center for Laboratory Animals. Male *m**ecp2*
^null/y^ mice with a C57BL/6J background were generated for >10 generations [53]. All behavioral assessments were performed by a rater who was blinded to the genotype. All animals were kept on a 12:12 h light–dark cycle at 22 °C with food and water available ad libitum. All experimental procedures were performed during the light-on phase of the cycle.

### 4.2. Music Treatment

Wild-type C57BL/6J (WT) and *mecp2*
^null/y^ mice were weaned 24 days after birth. After genotyping, male mice were selected for the following study. At P25, animals were randomly divided into six groups: (1) control wild-type mice (C-WT); (2) the wild-type white noise group (W-WT), in which wild-type mice received 50–60 dB white noise at a frequency of 300–10,000 Hz; (3) the wild-type music-based intervention group (M-WT), in which wild-type mice received a music-based intervention involving 50–60 dB music at a frequency of 300–10,000 Hz; (4) control *mecp2 ^null/y^* mice (C-*mecp2 ^null/y^*); (5) the *mecp2 ^null/y^* white noise group (W-*mecp2 ^null/y^*), in which the *mecp2 ^null/y^* mice received 50–60 dB white noise at a frequency of 300–10,000 Hz; and (6) the *mecp2 ^null/y^* music-based intervention group (M-*mecp2 ^null/y^* mice), in which *mecp2* ^null/y^ received a music-based intervention involving 50–60 dB music at a frequency of 300–10,000 Hz. Animals were exposed to music or white noise for 6 h/day for 3 consecutive weeks. The distance between mouse cages and the sound box was 1 m. All experiments were performed in a quiet environment to avoid interference from external noise. After 3 weeks in lit conditions, animal behaviors, including locomotion, three-chamber social ability, and anxiety-like behaviors, were assessed. At the end of the experiments, mice were killed, and four brain regions, the prefrontal cortex (PFC), hippocampus, basal ganglia, and amygdala, were dissected from fresh brain tissue (see Appendix A).

### 4.3. Music Selection

Six compositions containing different music forms were chosen for use in this study. These included vocal and instrumental musical items. First, the classic Taiwanese melody “Rainy Night Flower”, played by Evergreen symphony, was chosen. Second, we chose classical music: piano concerto Mozart Sonata for 2 pianos K448, which has been reported to be effective for treating refractory epilepsy [54,55]. Third, we used vocal music: solo compositions of Sarah Brightman, whose voice is delicate and sweet. Fourth, music sung by a church choir was selected because of its harmonious and solemn nature. Fifth, we selected preludes and fugues by Bach. Sixth, we chose a piano concerto by Chopin. Each melody line was enriched with harmony.

### 4.4. Open Field Test

An open field test was performed to probe the anxiety status following music exposure. Exploratory activity and anxiety-like behaviors were evaluated in an automated open field. The apparatus was set up under a digital camera, which was connected to a video recorder and a computer under the control of a Smart tracking system (TSE, Germany). Mice were placed in the center of the open field and left to explore freely for 10 min. Activity was assessed with a computer-assisted digital scan optical animal activity system, TSE multiconditioning system (TSE, Germany). The total distance moved over 10 min in the arena was recorded as a measure of locomotor activity. Time spent in a central square (20 × 20 cm^2^) in the open field during the first 10 min was automatically recorded as center time and was used as a measure of anxiety-like behavior [56].

### 4.5. Three-Chamber Social Ability and Social Novelty Measurement

Three-chamber social ability and social novelty measurements were performed to characterize cognition in terms of general social ability and novelty. Social ability and social novelty measures were conducted in a three-chamber cage as previously described with the following specifications: 60 × 40 × 22 cm (L × W × H) [57]. The chamber was separated into three parts: 21 cm left and 21 cm right compartments together with an 18 cm middle compartment. There were two cylinder chambers that were 15 cm in height and 10 cm in diameter in the left and right compartments. During the social ability test, a stranger mouse (stranger 1) was placed inside the cylinder in the left compartment, with the cylinder in the right compartment remaining empty. The experimental mouse was placed in the middle chamber for 10 min and the amounts of time spent sniffing stranger 1 and the empty chamber were recorded. After 10 min, the experimental mouse and stranger 1 were taken out and the chambers were cleaned. After a 10-min break, the experimental mouse and stranger 1 were placed back in their original chambers, and a second stranger mouse (stranger 2) was placed in the cylinder in the right compartment. The amounts of time spent sniffing stranger 1 and stranger 2 by the experimental mouse were recorded during a 10 min observation period for the social novelty test. Social interaction behaviour was also measured, and the specifications for the three-chamber cage were 108 × 50 × 42 cm (L × W × H).

### 4.6. Quantitative Reverse Transcription (qRT-PCR) for BDNF mRNA Expression

Total RNA was extracted from fresh tissues using the TRIzol protocol (Invitrogen, San Diego, CA, USA) [58]. We assessed RNA purity and integrity with OD260/OD280 spectrophotometric measurements. A 5 µg portion of total RNA and 1.5 µg of oligo-dT primers were incubated at 70 °C for 10 min and gradually cooled to room temperature. Each RT mixture containing 25 units of M-MLV reverse transcriptase (Promega, Madison, WI, USA), 10 L 5 × reaction buffer, 0.5 mM dNTP, and nuclease-free distilled water was added to a final volume of 50 L. The amount of cDNA was quantified using the LightCycler SYBR-Green 1 Master mix (Roche, Basel, Switzerland) by real-time PCR. For normalization, Actin cDNA levels were analyzed. Ct values from each sample were obtained using Light Cycler 480 software (Roche, Basel, Switzerland). The samples were incubated at 37 °C for 90 min followed by enzyme denaturation at 95 °C for 10 min. Each PCR (20 L) contained 2 µL of RT product, 1 unit of Taq DNA polymerase (Viogene, Taipei, Taiwan), 2 µL of 10 × PCR buffer plusMgCl2, 0.2 mM of dNTP, and 0.5 M of gene-specific primers (*BDNF*: forward5′-GACAAGGCAACTTGGCCTAC-3′, reverse 5′-CTGTCACACACGCTCAGCTC-3′; *FNDC5*: forward, 5′-CTCTCTCTTGGCTTCTCTCTTTC-3′ reverse 5′-CATGGACATTGCTGAGGTACT-3′; Actin: 5′-AGGCCAACCGTGAAAAGATG-3′ and 5′-TGTGGTACGACCAGAGGCATAC-3′). The amplified reaction was performed using a thermocycler for a single 3 min initial denaturation period at 94 °C followed by 30 cycles of *BDNF*, 30 cycles of *FNDC5*, and 20 cycles of Actin under the following conditions: 94 °C (20 s), 53 °C (20 s), 72 °C (20 s), and a final extension at 72 °C for 4 min.

### 4.7. Western Blotting Assessment

Fresh tissues in the prefrontal cortex, hippocampus, basal ganglia, and amygdala were selected for Western blotting. The brain regions were rapidly dissected. We homogenized the protein in a cell lysis buffer (Fermentas, Burlington, ON, Canada), containing a complete protease inhibitor cocktail (Thermo, Rockford, AL, USA). Subsquently, 12% sodium dodecyl sulphate-polyacrylamide gel was used to separate equal amounts of protein from each sample, and these were electrophoretically transferred onto polyvinylidene fluoride membranes (Millipore, Burlington, MA, USA). The membranes were incubated with rabbit anti-*BDNF* (1:5000; Abcam, 15 kDa) and TrkB (1:5000, Abcam, 145/95kDa) overnight at 4 ℃ after blocking with 5% skimmed milk in Tris-buffered saline with 0.1% Tween 20 (TBST) for 1 h at 25 ℃. The membranes were then washed in TBST and incubated for 1 h at room temperature with horseradish peroxidase-conjugated secondary antibody (1:10,000; Abcam, Cambridge, UK). Immunoreactivity was visualized with the enhanced chemiluminescence method.

### 4.8. Statistical Analyses

Two-sample comparisons were performed using the Student’s *t* test, and multiple comparisons were performed using the one-way ANOVA and the Tukey post hoc test. Statistical analyses were carried out using GraphPad Prism v.6.0 (GraphPad Software, San Diego, CA, USA). All data are presented as the mean ± SEM or with box plot diagrams, and statistical significance was accepted at the 5% level.

## 5. Conclusions

Our results provided evidence that although music-based interventions do not ameliorate motor deficits and mitigate anxiety in *mecp2 ^null/y^* mice, they can be used to reduce social function impairment in *mecp2 ^null/y^* mice. The effects of music-based interventions occur by upregulating *BDNF* gene expression without activating the TrkB receptor pathway through modulation of the *FNDC5* gene in the PFC and hippocampal region.

## Figures and Tables

**Figure 1 ijms-22-07174-f001:**
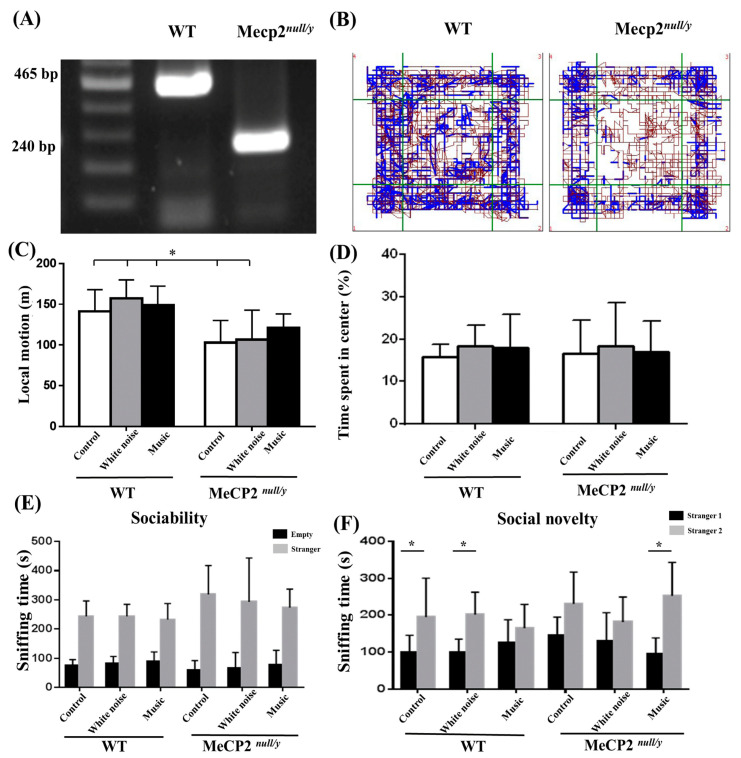
Motor function and anxiety-related behaviors tested in the open field and social ability tested with three-chamber social ability and social novelty measurements. (**A**) Genomic tail DNA was prepared for all animals before the experiment. (**B**) The representative movement tracks of wild-type mice (left) and *mecp2 ^null/y^* mice (right) in the open field are shown. (**C**) The traveling distances were significantly lower for C-*mecp2 ^null/y^* and W-*mecp2 ^null/y^* mice as compared with all wild-type mice. (**D**) The pecentage of time spent in the central zone was comparable for *mecp2 ^null/y^* and wild-type mice. (**E**) No significant differences in the sniffing of stranger mice between *Mecp2 ^null/y^* and wild-type mice were identified in the social ability test. (**F**) The M-*mecp2 ^null/y^* mice spent more time sniffing novel mice as compared with the C-*mecp2 ^null/y^* mice. (* *p <* 0.05).

**Figure 2 ijms-22-07174-f002:**
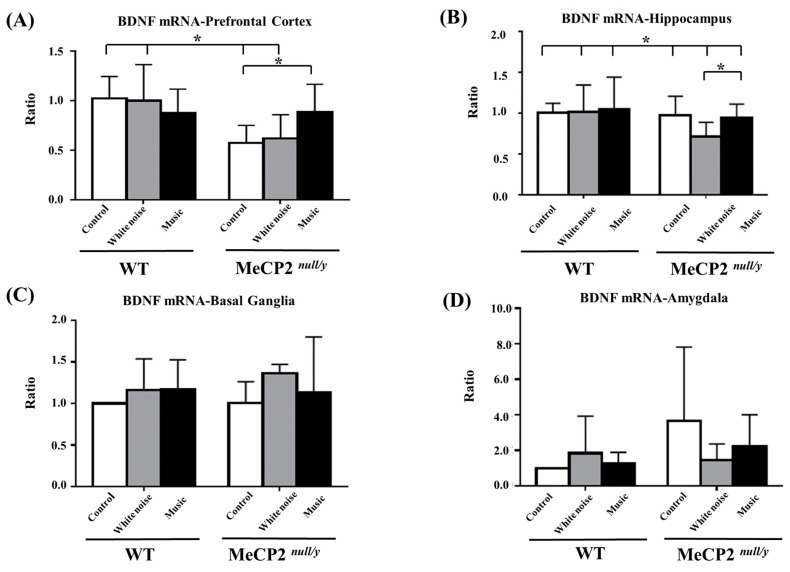
Comparison of *BDNF* mRNA expression between wild-type mice and *mecp2 ^null/y^* mice in different brain regions. (**A**) Prefrontal cortex, (**B**) hippocampus, (**C**) basal ganglia, (**D**) amygdala (* *p*
*<* 0.05).

**Figure 3 ijms-22-07174-f003:**
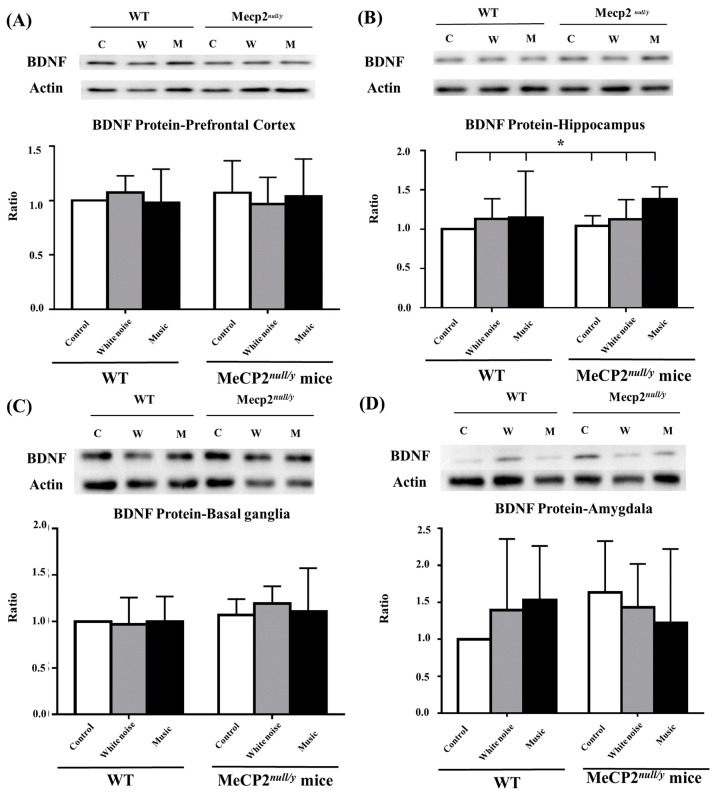
Comparison of the *BDNF* protein level in wild-type mice and *mecp2 ^null/y^* mice in different brain regions. (**A**) Prefrontal cortex, (**B**) hippocampus, (**C**) basal ganglia, (**D**) amygdala (* *p*
*<* 0.05).

**Figure 4 ijms-22-07174-f004:**
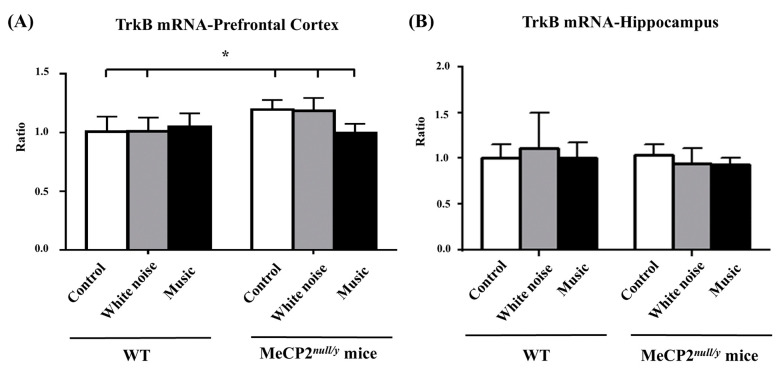
Comparison of TrkB mRNA expression between wild-type mice and *mecp2 ^null/y^* mice in the PFC and hippocampus. (**A**) *Mecp2 ^null/y^* mice that did not undergo a music-based intervention showed significantly increased TrkB mRNA expression in the PFC as compared with all wild-type mice and *mecp2 ^null/y^* mice that underwent a music-based intervention. (**B**) TrkB mRNA expression was comparable in the hippocampus between wild-type mice and *mecp2 ^null/y^* mice. (* *p <* 0.05).

**Figure 5 ijms-22-07174-f005:**
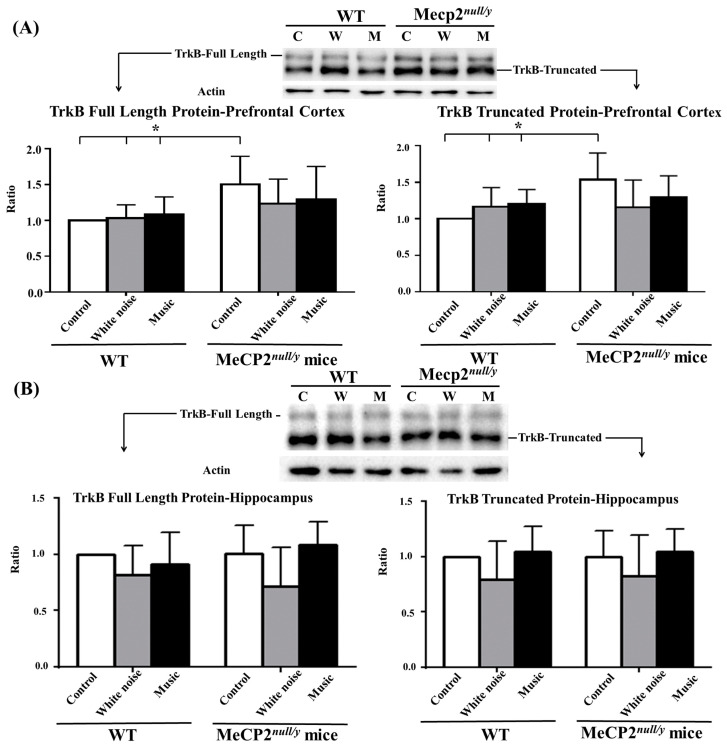
Analysis of full length and truncated TrkB proteins in wild-type and *mecp2 ^null/y^* mice in the PFC and hippocampus. *Mecp2 ^null/y^* mice that did not undergo a music-based intervention produced significantly higher full length and truncated TrkB protein levels in the prefrontal cortex (**A**) but not in the hippocampus (**B**) as compared with all wild-type mice and *mecp2 ^null/y^* mice that underwent a music-based intervention (* *p <* 0.05).

**Figure 6 ijms-22-07174-f006:**
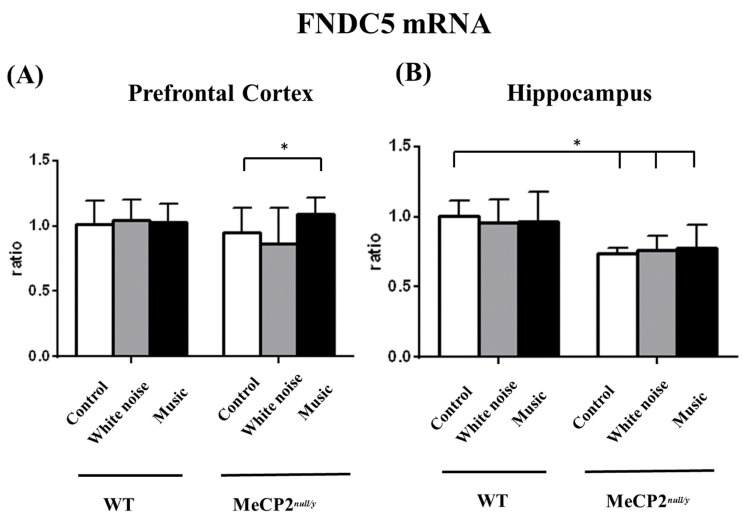
Comparison of *FNDC5* expression in the PFC and hippocampus between wild-type and *mecp2 ^null/y^* mice. (**A**) *Mecp2 ^null/y^* mice that underwent a music-based intervention showed significantly increased *FNDC5* expression in the PFC as compared with control *mecp2 ^null/y^* mice. (**B**) *Mecp2 ^null/y^* mice had significantly lower *FNDC5* expression as compared with all wild-type mice (* *p* < 0.05).

## Data Availability

Not applicable.

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
