# Peer review of "Music-Based Intervention Ameliorates Mecp2-Loss-Mediated Sociability Repression in Mice through the Prefrontal Cortex FNDC5/BDNF Pathway"

_ijms, 2021, doi:10.3390/ijms22137174_

Round 1

Reviewer 1 Report

Thank you for inviting me to review this interesting paper entitled "Music Therapy Ameliorates Mecp2 loss-Mediated Sociability Repression in Mice through Prefrontal Cortex FNDC5/BDNF Pathway" which described the effect of a music intervention of a mouse model of Rett syndrome. The paper is well-written and the procedures clearly described. I have only a couple of minor considerations that may help the authors improve the paper:

  • It has been shown that BDNF is actually increased in children with ASD (see for instance doi: 10.1038/s41598-020-79080-w) as well as associated with autistic traits in the general population (doi: 10.1038/s41598-020-79080-w). ASD, analogously to Rett's syndrome, is characterized by impairments in socialization. How can the authors explain the discrepancy? Please, discuss appropriately.
  • I have a perplexity about the use of the expression "music therapy". Music therapy implies the presence of a therapist in the setting (which is not possible in the case of animal models), or at least the involvement of a therapist in the choice of the music. Was the music chosen by a therapist? Was there a rationale for the choice of particular music? If not, I think it would be better to talk more generically about "music" generically or "music-based intervention"  

Author Response

#Comments from reviewer1

Thank you for inviting me to review this interesting paper entitled "Music Therapy Ameliorates Mecp2 loss-Mediated Sociability Repression in Mice through Prefrontal Cortex FNDC5/BDNF Pathway" which described the effect of a music intervention of a mouse model of Rett syndrome. The paper is well-written and the procedures clearly described. I have only a couple of minor considerations that may help the authors improve the paper:

Point 1. It has been shown that BDNF is actually increased in children with ASD (see for instance doi: 10.1038/s41598-020-79080-w) as well as associated with autistic traits in the general population (doi: 10.1038/s41598-020-79080-w). ASD, analogously to Rett's syndrome, is characterized by impairments in socialization. How can the authors explain the discrepancy? Please, discuss appropriately.

Respond 1. I am very appreciated of reviewer’s comments. Indeed, autistic spectrum disorder (ASD) and RTT are neurodevelopmental disorders sharing common features, including limited eye contact and poor socialization, however, clear differences between two disorders exist in the aspects of clinical presentations and neurotrophin signaling. As for the aspect of clinical considerations, Autism occurs in males predominantly, and it usually lacks of genetic mutation. ASD is associated with accelerated rate of head growth. In addition, ASD had a greater incidence that RTT (1:100 v.s. 1:1500 female births). Otherwise, RTT is a X-lined neurodevelopmental disorder, involving MeCp2 gene mutation so that females are mostly affected. RTT is found postnatal deceleration rate of head growth. As for the molecular mechanism of neurotrophin signaling, patients with RTT have reported decreased blood and cerebrospinal fluid (CSF) levels of BDNF (Katz, 2014), wherease patients with BDNF level was found significantly elevated in patients with ASD (Nishimura et al.,2017; Sadakata et al.,2012; Ricci et al.,2013). I had re-wrote the sentences in the second paragraph of introduction and inserted the references (doi: 10.1038/s41598-020-79080-w).

Point 2. I have a perplexity about the use of the expression "music therapy". Music therapy implies the presence of a therapist in the setting (which is not possible in the case of animal models), or at least the involvement of a therapist in the choice of the music. Was the music chosen by a therapist? Was there a rationale for the choice of particular music? If not, I think it would be better to talk more generically about "music" generically or "music-based intervention"  

Respond 2. Thanks reviewer for providing me such a wonderful recommendation! We did not set a music therapist for choosing music in this study. We chose the music based on previous publication instead of choosing music with particular pitch, rhythm or melody (Lin et al, 2011; Bedetti et al, 2019), so that I re-wrote “ music-based intervention” instead of “ music therapy” in the manuscript. 

Comments from reviewer 2

In this research article, the authors did an evaluation of the effect of music therapy using MeCP2-null mice model. It was observed improvements in social deficits upon exposure to music therapeutics, which is a feature associated with RTT syndrome. Interestingly, within this study it was observed that the mechanism behind the effect of music therapy were associated with the upregulation of BDNF mRNA expression in the prefrontal cortex, through the modulation of FNDC5 gene. The topic falls within the scope of journal and reveals to be an interesting and innovative field of study to be further explored in RTT pathology. I would like to suggest the authors to improve some details and to discuss some more information, before considering appropriate for publication. Please see the comments.

 Point 1. In the introduction section, it would be interesting if the authors could explore deeper the mechanisms, possible pathways and gene patterns altered in brain/specific regions, after music therapeutics. The authors mentioned alterations in synaptic plasticity and BNDF/TrkB signaling, which is absolutely correct and in context, but some more detailed information could be added. I would like to suggest this publication (https://doi.org/10.1016/j.brainresbull.2009.05.020), among others.

Respond 1. Thank you for your suggestion. I added the sentences in the third paragraph of introduction: “Previous studies reported that BDNF, NGF levels [22] and some genes were regulated by microarray after music exposure [23]”. The suggested publication (https://doi.org/10.1016/j.brainresbull.2009.05.020) was inserted as reference 23.

Point 2. The quality of the figures are quite low (all of them). The authors should increase the quality, by including clearer images with high quality, at least 300 DPI.

Respond 2. Thank you for reviewing the manuscript in details. All the figures had been re-edited with high quality at least 300 DPI. All of them were presented in the manuscript as well as uploaded alone in Zip file. All the newly re-edited figures had been re-submitted.

Point 3. The authors wrote a very interesting and complete discussion, focusing not only on the main results obtained, but also on the limitations of the model. I would only like to suggest the addition of a brief paragraph discussing the possibility of testing the effect of music therapy (associated with the effects of distinct frequencies) using in vitro models, such as brain derived neuronal hiPSCs-cells or cortical organoid models. Since these models could complement the data observed in vivo, focusing more into the molecular pathways and also in the possibility of being used for functional analysis, upon exposure to the different musical conditions (distinct musical frequencies/ HZ). I would like to suggest you to include these two recent papers 3389/fcell.2020.610427 and https://doi.org/10.15252/emmm.202012523.

 Respond 3. Thank you so much for providing me the new engineering biotech which may be helpful for my future study. I had added a brief paragraph about neuronal hiPSCs-cells or cortical organoid models in RTT in vitro model in the fifth paragraph of discussion, and the suggested papers were cited as reference 55 and reference 56. 

Reviewer 2 Report

In this research article, the authors did an evaluation of the effect of music therapy using MeCP2-null mice model. It was observed improvements in social deficits upon exposure to music therapeutics, which is a feature associated with RTT syndrome. Interestingly, within this study it was observed that the mechanism behind the effect of music therapy were associated with the upregulation of BDNF mRNA expression in the prefrontal cortex, through the modulation of FNDC5 gene. The topic falls within the scope of journal and reveals to be an interesting and innovative field of study to be further explored in RTT pathology. I would like to suggest the authors to improve some details and to discuss some more information, before considering appropriate for publication. Please see the comments.

  1. In the introduction section, it would be interesting if the authors could explore deeper the mechanisms, possible pathways and gene patterns altered in brain/specific regions, after music therapeutics. The authors mentioned alterations in synaptic plasticity and BNDF/TrkB signaling, which is absolutely correct and in context, but some more detailed information could be added. I would like to suggest this publication (https://doi.org/10.1016/j.brainresbull.2009.05.020), among others.
  2. The quality of the figures are quite low (all of them). The authors should increase the quality, by including clearer images with high quality, at least 300 DPI.
  3. The authors wrote a very interesting and complete discussion, focusing not only on the main results obtained, but also on the limitations of the model. I would only like to suggest the addition of a brief paragraph discussing the possibility of testing the effect of music therapy (associated with the effects of distinct frequencies) using in vitro models, such as brain derived neuronal hiPSCs-cells or cortical organoid models. Since these models could complement the data observed in vivo, focusing more into the molecular pathways and also in the possibility of being used for functional analysis, upon exposure to the different musical conditions (distinct musical frequencies/ HZ). I would like to suggest you to include these two recent papers 3389/fcell.2020.610427 and https://doi.org/10.15252/emmm.202012523.

Author Response

Comments from reviewer 2

In this research article, the authors did an evaluation of the effect of music therapy using MeCP2-null mice model. It was observed improvements in social deficits upon exposure to music therapeutics, which is a feature associated with RTT syndrome. Interestingly, within this study it was observed that the mechanism behind the effect of music therapy were associated with the upregulation of BDNF mRNA expression in the prefrontal cortex, through the modulation of FNDC5 gene. The topic falls within the scope of journal and reveals to be an interesting and innovative field of study to be further explored in RTT pathology. I would like to suggest the authors to improve some details and to discuss some more information, before considering appropriate for publication. Please see the comments.

 Point 1. In the introduction section, it would be interesting if the authors could explore deeper the mechanisms, possible pathways and gene patterns altered in brain/specific regions, after music therapeutics. The authors mentioned alterations in synaptic plasticity and BNDF/TrkB signaling, which is absolutely correct and in context, but some more detailed information could be added. I would like to suggest this publication (https://doi.org/10.1016/j.brainresbull.2009.05.020), among others.

Respond 1. Thank you for your suggestion. I added the sentences in the third paragraph of introduction: “Previous studies reported that BDNF, NGF levels [22] and some genes were regulated by microarray after music exposure [23]”. The suggested publication (https://doi.org/10.1016/j.brainresbull.2009.05.020) was inserted as reference 23.

Point 2. The quality of the figures are quite low (all of them). The authors should increase the quality, by including clearer images with high quality, at least 300 DPI.

Respond 2. Thank you for reviewing the manuscript in details. All the figures had been re-edited with high quality at least 300 DPI. All of them were presented in the manuscript as well as uploaded alone in Zip file. All the newly re-edited figures had been re-submitted.

Point 3. The authors wrote a very interesting and complete discussion, focusing not only on the main results obtained, but also on the limitations of the model. I would only like to suggest the addition of a brief paragraph discussing the possibility of testing the effect of music therapy (associated with the effects of distinct frequencies) using in vitro models, such as brain derived neuronal hiPSCs-cells or cortical organoid models. Since these models could complement the data observed in vivo, focusing more into the molecular pathways and also in the possibility of being used for functional analysis, upon exposure to the different musical conditions (distinct musical frequencies/ HZ). I would like to suggest you to include these two recent papers 3389/fcell.2020.610427 and https://doi.org/10.15252/emmm.202012523.

 Respond 3. Thank you so much for providing me the new engineering biotech which may be helpful for my future study. I had added a brief paragraph about neuronal hiPSCs-cells or cortical organoid models in RTT in vitro model in the fifth paragraph of discussion, and the suggested papers were cited as reference 55 and reference 56.